# Radiotherapy and Immunotherapy, Combined Treatment for Unresectable Mucosal Melanoma with Vaginal Origin

**Laura Rebegea** [1,2,3]🆔, **Dorel Firescu** [4], **Gabriela Stoleriu** [2,5,*]🆔, **Manuela Arbune** [2]🆔, **Rodica Anghel** [6], **Mihaela Dumitru** [1], **Raul Mihailov** [2], **Anca Iulia Neagu** [6] **and Xenia Bacinschi** [7]

1   Department of Radiotherapy, 'St. Apostol Andrei' Emergency Clinical Hospital, 800578 Galati, Romania; mihaeladumitru11@yahoo.com
2   Clinical Department, Faculty of Medicine and Pharmacy, 'Dunarea de Jos' University, 800010 Galati, Romania; laura_rebegea@yahoo.com (L.R.); arbunemanuela@yahoo.com (M.A.); raulmihailov@yahoo.com (R.M.)
3   Research Center in the Field of Medical and Pharmaceutical Sciences, ReFORM-UDJ, 'Dunarea de Jos' University, 800010 Galati, Romania
4   Surgical Department, Faculty of Medicine and Pharmacy, 'Dunarea de Jos' University, 800010 Galati, Romania; dorel.firescu@ugal.ro
5   'St. Spiridon' Hospital, 700111 Iași, Romania
6   Department of Morphological and Functional Sciences, Faculty of Medicine and Pharmacy, 'Dunarea de Jos' University, 800010 Galati, Romania; rodicamanghel@gmail.com (R.A.); ancazanoschi@gmail.com (A.I.N.)
7   8th Department, Faculty of Medicine, 'Carol Davila' University of Medicine and Pharmacy, 050474 Bucharest, Romania; xenia_bacinschi@yahoo.com
*   Correspondence: stoleriugabriela@yahoo.com; Tel.: +40-721688422

**Abstract:** Gynecologic melanomas are uncommon and malignant mucosal melanomas with vaginal origin are extremely rare, treatment strategies are limited and extrapolated from those of cutaneous melanoma. A better understanding of the vulvovaginal melanoma's biology and its risk factors is needed. Therapeutic strategies include surgery, systemic therapy and radiotherapy. For vulvovaginal melanoma, surgery is selected as the primary treatment. Immunotherapy and target treatment have recently enhanced the systemic therapy for cutaneous melanoma (CM). Immunotherapy and new target agents demonstrated a better survival of melanoma and might be considered as treatment of vulvovaginal melanoma. Radiotherapy is included in the therapeutic arsenal for mucosal melanoma and may be performed on selected patients who may receive concurrent checkpoints and inhibition neoadjuvant radiotherapy with the purpose of reducing morbidity and mortality.

**Keywords:** gynecologic melanomas; mucosal melanomas; vulvovaginal melanomas; immunotherapy; radiotherapy





## 1. Introduction

Malignant mucosal melanomas with vaginal origin are extremely rare. Data regarding immunotherapy and radiotherapy or combined treatment for mucosal melanoma (MM) are scarcely reported in the specialty literature. Due to the lack of therapeutic guides or protocols and due to the small number of clinical trials involving MM, the treatment approaches are extrapolated from cutaneous melanoma's therapeutic guides.

Gynecologic melanomas (GM) are rare, being 1% of all melanomas and accounting for 18% of mucosal melanoma (MM). The most frequent sites of GM are, in order, vulva, vagina and, less frequent, the cervix. The prognosis for gynecologic melanomas is poor, with overall survival (OS) at 5 years less than 50% for vulvar, less than 30% for vaginal situs and even smaller for cervix melanoma [1–3]. Melanoma has a noticeably increasing incidence. In 2016, 2.5 per 100,000 (9003 patients) died of melanoma in the United States [4,5]. Due to the rarity of lower genital tract MM, the information related to treatment considerations is few.

Surveillance Epidemiology and End Results (SEER) database of the U.S. National Cancer Institute (NCI) reported for over a 30-year period (1973–2003) that only 644 cases of vulvar melanoma were identified and most patients (85%) who presented vulvar melanoma were Caucasian. The widest published report from one institution of vaginal melanoma reported 37 cases only in a period of 29 years [6,7]. In 2003, the Annual Meeting of the Society of Gynaecologic Oncology from Memorial Sloan Kettering, over a long period registered fewer than 80 vulvar and fewer than 45 vaginal melanomas [6,8].

It is known that melanoma is a very immunogenic tumor. Thus, reverted radiotherapy can be used as an immunity-stimulating method during immunotherapy, making it more efficient. Until recently, it was considered to be a palliative treatment, yet the most recent cases proved it to be an adjuvant.

## 2. Materials and Methods

We report a case of diagnosed vaginal melanoma in the Oncology Department managed by the Oncology, Pathology and Radiotherapy Departments of 'St. Apostol Andrei' Emergency Clinical Hospital, Galati, Romania.

Immunotherapy started in January 2021 with nivolumab, 240 mg/2 weeks and the 3D conformal radiotherapy technique was performed 4 months later, in total a dose of 30 Gy, with good compliance. The patient was reevaluated by clinical and imaging examinations, and at 2 months after the end of radiotherapy, a partial remission conforming to the RECIST (Response Evaluation Criteria in Solid Tumors) criteria was observed.

## 3. Results

In our institution, we had to manage a case with vaginal melanoma. The patient was 70 years old and was diagnosed in June 2020 with the previously mentioned medical condition. In June 2020, the biopsy and imagistic evaluation was carried out. Head, neck, thorax, abdomen CT did not reveal metastatic disease in July 2020, but an MRI showed a vaginal tumor on the left side wall, of 50/32/27 mm in size, without lymph node invasion. In September 2020, the removal of the vaginal tumor (left lateral wall) was performed with limited data regarding resection margins. The initial histopathological exam revealed undifferentiated carcinoma and the immunohistochemical tests showed that tumor cells were diffusely positive for S100; while HMB45 and p16 were positive with moderate intensity in the tumor cells; p63 and CK5/6 were negative in the tumor and the Ki-67 labeling index was approximately 60% (Figures 1 and 2). Thus, the diagnosis of an ulcerated nodular malignant melanoma was established.

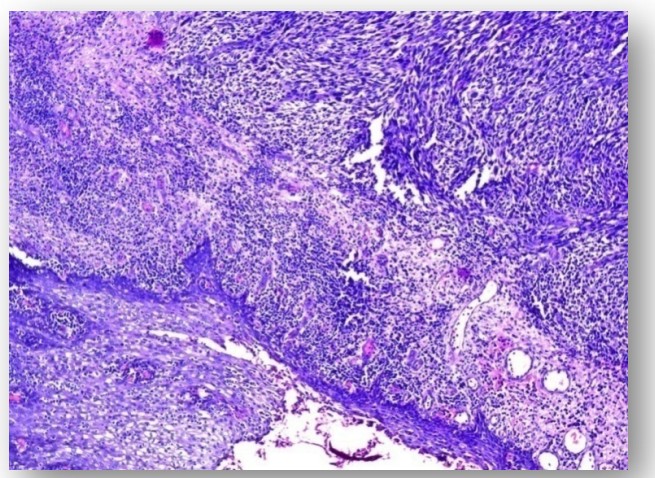

**Figure 1.** The squamous cell epithelium (in the lower left quadrant), mononuclear cell infiltrate in the subepithelial connective tissue and the tumor proliferation, consisted of sheets of spindle cells, H&E stain, and magnification of ×100.

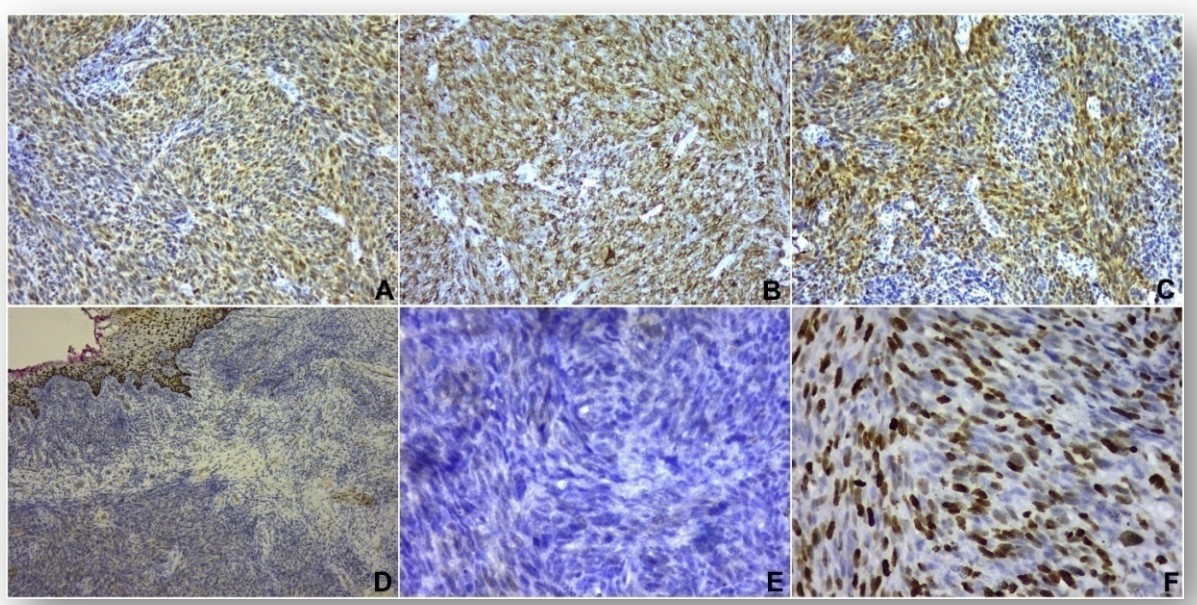

**Figure 2.** Immunohistochemical profile of the tumor: (**A**) Tumor cells are diffusely positive for S100, magnification ×200; (**B**) The tumor shows intense, diffuse positivity for HMB45, magnification ×200; (**C**) p16 is positive with moderate intensity, magnification ×200; (**D**) p63 is negative in the tumor cell and positive in the squamous epithelium (left upper quadrant), magnification ×100; (**E**) Tumor cells show negativity for CK5/6, magnification ×400; (**F**) A Ki-67 labeling index of 60%, magnification ×400.

Imagistic evaluation by computed tomography evidenced a lung node with a diameter of 3 mm and a left inguino-femoral lymphadenopathy with a diameter of 35 mm in October 2020. Surgery was performed, consisting of an inguino-femoral lymphadenectomy with the excision of the adenopathy block, in November 2020. The final histopathological examination after the paraffin-embedding of the tissue determined that one of the seven examined lymph nodes had a small area of malignant tumor infiltration, with associated chronic inflammatory infiltrate. Biological, biochemical parameters and tumor markers were within the normal limits. Additionally, the BRAF mutation test was negative.

One month after the surgery, a positron emission computed tomography examination was performed and bilateral lung nodes, with SUV 18 mm were revealed.

Our oncology committee decided upon immunotherapy performed with nivolumab, which was initiated in January 2021, conforming to national protocols. Radiotherapy was integrated into pluridisciplinary care and was performed in our department, in 3D conformal technique with 6 MV photons, the total dose of 30 Gy in 10 fractions, was applied with good compliance. Organ at risk dose constraints were evaluated conforming to QUANTEC (Quantitative Analysis of Normal Tissue Effects in the Clinic) (Figure 3) and hospital therapeutic protocols, after the informed consent of the patient was signed.

Twelve months after immunotherapy was initiated, the patient had a performance index of IP (ECOG) = 1, representing a complete remission of lung lesions without immune toxicities.

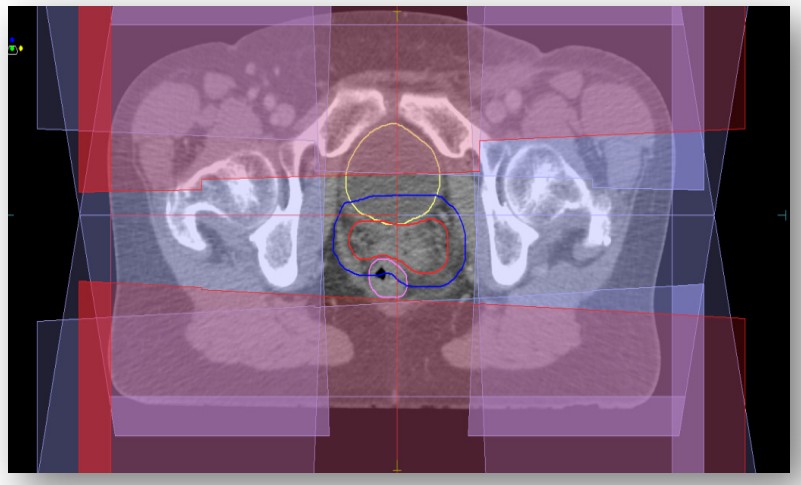 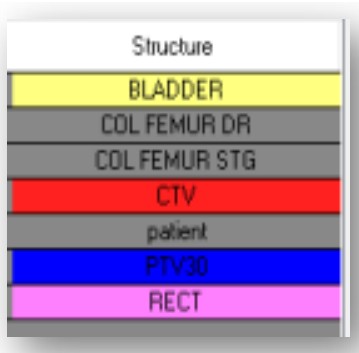

(**a**)                                                                (**b**)

**Figure 3.** Aspects of radiotherapy plan highlighting the structures: (**a**) the description of field distribution; (**b**) the description of structures defined in planned treatment.

## 4. Discussions

Given the low incidence and rarity of gynecological mucosal melanoma being reported, most of the data regarding patient care and treatment have been deduced from other studies that included cutaneous and mucosal melanomas of diverse origins. Primary surgical resection represents the first therapeutic option and the scope of this is to obtain negative margins [9,10]. Different melanoma subtypes have distinct molecular lines. Mucosal, acral and chronic sun-induced damage (CSD)—associated melanomas have different characteristics of chromosomal damage in contrast with those appearing from skin without CSD [11,12].

Regarding biological characteristics, MM and cutaneous melanoma (CM) are different. Thus, MMs have a significantly smaller percentage, less than 15% of BRAF gene mutations, than CM without chronic sun damage [1,13–15]. Moreover, the increased BRAF gene mutation in MM damages gene regions other than codon 600 or is a non-activating mutation, so it is not expected to have an effect on target therapy [1,14]. Moreover, gene copy numbers and structural changes, for example in KIT, are significantly more magnified in MM than in CM [1,16,17].

The treatment of MM involves a multidisciplinary team, as it is a multimodal therapy adapted to disease stage. In localized disease instances, surgery is the main treatment followed by adjuvant therapy in stage 3 of the disease; adjuvant radiotherapy is indicated in clinical positive lymph nodes. In metastatic disease, target treatment and immunotherapy are the principal approaches which can integrate radiotherapy and surgery in well selected cases [3,18].

For vulvovaginal melanoma, surgery, if it is possible, is to be preferred as primary treatment, not requiring an exenterative procedure.

Surgery has two important scopes: the treatment of the primary tumor and the assessment of lymph node regions. Surgical cytoreduction can be performed in individualized in well selected metastatic cases and is not a routine practice [6,19].

For newly diagnosed vulvovaginal MM or for a relapsed disease, pelvic exenteration is not usually performed.

Sentinel lymph node (SLN) mapping of the inguinal lymph nodes can be easily carried out by experienced surgeons, but strong recommendations regarding this technique in vaginal melanomas cannot be given [6,20] and the appropriate approach of SLN-positive

cases and the role of additional inguino-femoral lymphadenectomy needs to be determined in further studies [6].

The therapeutic value of SLN biopsy is limited, being more important as a tool for diagnosis. SLN biopsy improved disease-free survival by 7% and 10% for patients with intermediary thickness (1.2–3.5 mm diameter) or thick (3.5 mm diameter) primary lesions [5].

The ongoing discussions over the importance of lymph node dissection will end when molecularly guided imaging or a new biological therapy becomes accessible intended to identify and treat metastatic disease [21].

Immunotherapy and target therapy has demonstrated strong efficacy in the treatment of CM [9]. It has been reported that novel immunotherapeutic and targeted agents increase survival in melanoma and should be considered in cases of vulvovaginal melanoma. The use of vemurafenib in melanomas with a BRAFV600E mutation significantly increased progression-free survival (PFS) as opposed to dacarbazine in a phase III trial [22]. The median PFS was 5.3 months for vemurafenib against 1.6 months for dacarbazine.

BRAF inhibitor concurrent with MEK inhibitors is an elective procedure in the case of a BRAF V600-activating mutation [1]. If there is no mutation, anti-PD1 as unique therapy or combined therapy (nivolumab and ipilimumab) is recommendable [1,23].

The efficacy of associating nivolumab with ipilimumab seemed to be greater than the single use of the agents and the activity was lower in MM than in CM [1,16].

Other issue refers to the gene copy number and structural variations, such as in KIT, which are much more numerous in MM than in CM.

KIT inhibitors and tyrosine kinase inhibitors: TKIs important for the treatment of MM include Imatinib, Nilotinib, and Dasatinib [1]. In the last 4 years, many articles regarding the efficiency of immunotherapy with MM have been published, and we are mentioning here Hamid et al. [24] and Nathan et al. [25].

CheckMate 172 was a phase II, single-arm, open-label, multicenter study that evaluated nivolumab in melanoma patients with advanced disease who manifested relapsed disease during or after ipilimumab treatment.

The authors report data on 1008 treated patients, 6.3% of cases with mucosal melanoma, non-acral cutaneous melanoma in 71.7% of cases and acral cutaneous melanoma in 5.5% of cases. The minimum monitoring period was 18 months and median OS was 25.8 months for acral cutaneous melanoma, 25.3 months for non-acral cutaneous melanoma and 18-month OS rates were less than 60% for both. Median OS for ocular and mucosal melanoma was under 13 months and OS rates were under 35% for both their localizations [26].

So, nivolumab treatment after ipilimumab therapy is similar for melanoma subtypes and has a safety profile. OS was similar for both non-acral cutaneous and acral cutaneous melanoma situations. On the other hand, ocular and mucosal melanoma had differing median OS [25].

A study by Anko et al. [1] evaluated, in a retrospective study, five retrospective case series and nine case reports with gynecologic melanomas treated with immunotherapy (immune checkpoint inhibitor) and/or targeted therapy. This study is a comprehensive review of the literature data and randomized clinical trials regarding immunotherapy in mucosal melanoma. Among the articles reviewed by Anko et al., the same author published a study in 2020 that drew attention [26].

Daix et al. [27] published a study which described the management of one patient with gynecologic melanoma, originating in the vagina, regionally unresectable, with no primary treatment. The patient received nivolumab treatment, with complete response (CR), PFS of 8 months and OS of 8 months, being alive with no evidence of disease at the time of the article's publishing.

Inoue et al. [28] also published a study which described the management of one patient with gynecologic melanoma, originating in the vagina, with distant disease (brain) without primary systemic treatment, performed immunotherapy with nivolumab, with confirmed progressive disease (CPD) and PFS for 2 months. The OS was not specified, but the patient was alive with disease at the time the article was published.

Although for a long time, malignant melanoma was considered radioresistant, radiotherapy may play a role in vulvovaginal melanoma management, especially in the domain of symptom control. There are limited data to provide recommendations, but the use of hypofractionated radiation therapy may be considered to possibly achieve a less radical local excision.

Radiotherapy is applied using different advanced conformal techniques or radiosurgery (SRS) and it is part of the therapeutic arsenal of MM complications: spinal cord compression syndrome; brain secondary brain lesions; secondary bone lesions, in antalgic scope; an anti-hemorrhagic scope for obstruction management; metastases and extracranial disease. Radiotherapy is also used as a consolidation treatment for residual tumors. It was reported that control rates at one year range between 60 to 90% after SRS for melanoma brain metastases [29].

Regarding techniques, IMRT, IGRT or SRS are preferred due to the better sparing of organs that are at risk and fewer, later radiotherapy effects [5].

Vulvovaginal melanomas have an increased recurrence rate, higher than in other cutaneous/mucosal melanomas. Conforming to National Comprehensive Cancer Network (NCCN) [4], adjuvant radiotherapy may be applied in selected cases: radiotherapy regimens are not well established but include: Total Dose (TD) = 60–66 Gy in 30–33 fractions over 6–7 weeks, TD = 48 Gy in 20 fractions over 4 weeks and TD = 30 Gy/5 fractions over 2 weeks.

Definitive or palliative radiotherapy for regional metastases can also be given in TD = 50 Gy/20 fractions/4 weeks, TD = 30 Gy in 10 fractions/2 weeks, TD = 30 Gy in 5 fractions/2 weeks, TD = 20 Gy in 5 fractions/1 week and TD = 8 Gy in 1 fraction/1 day [4]. Hypofractionation appears to be equivalent in effectiveness to conventional fractionation with mild toxicity.

The combination of radiotherapy plus immunotherapy offers a chance to increase the immunostimulatory potential of radiation and appears to be a secure treatment, which is also sustained by strong biological reasons. Valid data confirm that radiotherapy is more usually used for metastatic rather than non-metastatic disease. Such a combination shows encouraging results in terms of survival results; however, further studies are needed to confirm such evidence [30].

It has been a long time since radiation was considered to determine immunogenic modulation; cell death, through the induction of dendritic cells; cell adhesion molecules; death receptors and tumor-associated antigens, as well as DNA strand breaks, apoptosis, and necrosis. In fact, radiation instead transforms the tumor into an individualized vaccine [29].

Due to the immune priming effect of radiotherapy, there is a great and complex biological motive and a strong proof for synergy in combination with immune checkpoint inhibitors, which these days are first-line therapy in patients with recurrent or metastatic melanoma.

Specialist information indicates that the bystander effect is induced by radiotherapy. This means the increasing therapeutic response of tumors outside of irradiated areas, in this way, also increases the possibility of tumor formation at distant sites [31].

Interaction between radiotherapy and immunotherapy was analyzed in retrospective trials and the results evidenced the abscopal response based on the activation of IFN-1 via cGAS and STING lane in the radio-treated neoplastic cells.

In a patient with a progressive disease during immunotherapy, radiotherapy was given for local control and very good systemic results were obtained [32]. There is a considerable potential to improve local control and abscopal effects by concurrent radiotherapy plus immunotherapy or radiotherapy plus hyperthermia or the combination of all three modalities, indicated as the next important trial in this refractory disease [30–32].

It is well known that there is a lack of data and clinical studies regarding combined radiotherapy with immunotherapy, yet recent studies show that some appropriate patients can perform concurrent neoadjuvant radiotherapy and immunotherapy to reduce the

toxicity of a planned surgical resection—which, historically, has a low chance for a systemic cure [9,31–35] and the efficacy of ipilimumab used concurrently with radiation in patients with metastatic melanoma highlights the potential synergy of this combination [33–37].

Schiavone et al. [9] presented a study with a few patients with vaginal and cervix melanoma. All four patients received ipilimumab with concurrent EBRT and three patients with persistent disease after combined treatment underwent less extensive surgical procedures. The radiation therapies doses varied between TD = 30 Gy/5 fractions and dose/fraction (d/fr) = 600 cGy with TD = 6020 cGy/28 fr, d/fr = 215 cGy.

Vaginectomy was performed during a period of between 33–97 days post-radiation therapy. All the information from clinic work and from the literature sustain the idea that palliative radiotherapy for secondary brain malignancy, also functioned for extracranial metastatic sites. In cases with increased risk of local and nodal relapse, the adjuvant radiotherapy is also recommended [29].

Gynecologic melanomas are rare and have bad prognostics. Due to the rarity of vulvo-vaginal mucosal melanoma, the vast majority of treatment-related data has been extracted from larger studies that include cutaneous malignant melanomas with varied origins.

We believe that future research must be focused not only on secondary metastatic lesions but also on a good palliative result of the primary lesion, and it is important to include these patients in clinical trials.

## 5. Conclusions

There are currently no practice guidelines for primary malignant melanomas located in the genital mucosa due to the rarity of this location. We considered it proper to apply the protocol for cutaneous melanoma and to integrate radiotherapy with immunotherapy in the interdisciplinary treatment of genital malignant melanoma located in the mucous membranes, which was a real success in this case without BRAF mutation.

The presence of a melanoma expert oncologist in the pluridisciplinary team, offers the optimal chance to cure and control the disease. Even if malignant melanoma was considered a radioresistant tumor, it was proven that hypofractionated radiotherapy schemes combined with immunotherapy have a beneficial effect in terms of survival and novel strategies for this neoplasia are required, knowing that immunotherapy administrated concomitantly with radiotherapy in cases with metastatic melanoma highlights the potential synergism of this association.

**Author Contributions:** Conceptualization, L.R. and X.B.; methodology, D.F. and M.A.; software, L.R., G.S. and X.B.; validation, L.R., G.S., R.A., M.D., A.I.N. and X.B.; formal analysis, D.F., R.M. and M.A.; investigation, R.A., M.D. and A.I.N.; resources, R.M., D.F., G.S. and M.A.; data curation, L.R., X.B., R.A., M.D. and A.I.N.; writing—original draft preparation, L.R.; writing—review and editing, L.R., X.B. and G.S.; visualization, L.R., D.F., M.A. and X.B.; supervision, X.B.; project administration, L.R. and X.B. All authors have read and agreed to the published version of the manuscript.

**Funding:** This research received no external funding.

**Institutional Review Board Statement:** The study was conducted in accordance with the Declaration of Helsinki and approved by the Ethics Committee of the 'St. Apostol Andrei' Emergency Clinical Hospital, Galati, Romania.

**Informed Consent Statement:** Written informed consent for participation in the study, histopathologic preparation of the tissues and for publication of this paper was obtained from the patient.

**Data Availability Statement:** The data presented in this study are available on reasonable request from the corresponding authors.

**Conflicts of Interest:** The authors declare no conflict of interest.

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
