# Peer review of "Radiotherapy and Immunotherapy, Combined Treatment for Unresectable Mucosal Melanoma with Vaginal Origin"

_applsci, doi:10.3390/app12157734_

Round 1

Reviewer 1 Report

In this rather short clinical paper, the authors present a single case of a 70 yrs old female patient with vaginal melanoma diagnosed in June 2020. The initial histopathological exam suggested undifferentiated carcinoma and the immunohistochemical tests showed that tumor cells were diffusely positive for S100, while HMB45 and p16 were positive with moderate intensity in the tumor cells, p63 and CK5/6 were negative in the tumor, and the Ki-67 labeling index was 85 approximately 60%. These are the molecular patterns nit really analyzed in the text. This is overal a potentially useful clinical application of combined immunotherapy and radiation therapy but the work needs major revisions. The abstract for example , conclusions and results are really non straight forward and highly informative. Details are missing. One reads and does not really understand which was the exact case and if the patient and other patients can really benefit from it. 

Materials and methods: very weak and without details. Radiation treatment details and the patient response needs to be better documented. 

In addition, a serious discussion must be made based on current knowledge incorporating papers documenting the role of the immune system in defining the overall outcome of radiation therapy :

1. Ventura, J.; Lobachevsky, P.N.; Palazzolo, J.S.; Forrester, H.; Haynes, N.M.; Ivashkevich, A.; Stevenson, A.W.; Hall, C.J.; Ntargaras, A.; Kotsaris, V., et al. Localized Synchrotron Irradiation of Mouse Skin Induces Persistent Systemic Genotoxic and Immune Responses. Cancer Res 2017, 77, 6389-6399, doi:10.1158/0008-5472.CAN-17-1066.

2. Rodriguez-Ruiz, M.E.; Vanpouille-Box, C.; Melero, I.; Formenti, S.C.; Demaria, S. Immunological Mechanisms Responsible for Radiation-Induced Abscopal Effect. Trends Immunol 2018, 39, 644-655, doi:10.1016/j.it.2018.06.001.

The text must be proofread for English and references have non-relevant numbering which shows not careful editing.

Author Response

Thank you very much for your professional evaluation, as well as your time and effort, in reviewing the article “Radiotherapy and Immunotherapy, combined treatment for unresectable mucosal melanoma with vaginal origin” - applsci-1767047 Rebegea et al, in order to be accepted for publication in MDPI Applied Sciences Journal, New Challenges in Skin Cancer  Special Issue.

According to your suggestions, we completed the missing data, Materials and methods details, remade and added the References, and corrected the spelling mistakes.

It is more clearly now: although malignant melanoma has been considered a radioresistant tumor, it was proven that hypofractionated radiotherapy schemes combined with immunotherapy have a beneficial effect in terms of survival and novel therapeutics and treatment strategies for this rare malignancy are needed, knowing that immunotherapy used concurrently with radiation in patients with metastatic melanoma highlights the potential synergy of this combination.

Reviewer 2 Report

Mucosal melanoma are rare and each treated patient increase the global knowledge on this topic. The discussion is very relevant and complete.

I have some concerns about some points:

Line 60: 2016???

In the “results” paragraph, could you describe also the pathologic stasus of margins and the microscopic depth of invasion? Moreover, at the diagnosis, do you make what type of surgery on vagina? Also the timing between surgery and immunotherapy and radiotherapy could be important and interesting to know, being today an increasing interest in synergism between radiotherapy and immunotherapy.

Line 142: please check if the reference 17 about uterine and extrauterine stromal sarcomas is relevant for mucosal melanoma treatment

Line 137-142: please provide references for treatments indications

Line 171: “vemuragfenib”, please correct

Line 214: “live”, please correct

Line 225: “or for metastatic disease”, I think it is a repetition, you have already described before the possible metastatic sites

Author Response

Thank you very much for your professional evaluation, as well as your time and effort, in reviewing the article “Radiotherapy and Immunotherapy, combined treatment for unresectable mucosal melanoma with vaginal origin” - applsci-1767047 Rebegea et al, in order to be accepted for publication in MDPI Applied Sciences Journal, New Challenges in Skin Cancer  Special Issue.

We would like to thank the third reviewer for his valuable ideas in correcting this manuscript.

Thank you for the appreciation of our work.

According to your suggestions, we have completed the proposed data, corrected the mistakes, and we hope you consider the article is based on current knowledge, documenting the role of the immune system in defining the overall outcome of radiation therapy combined with immunotherapy.

Round 2

Reviewer 1 Report

In this revised version it is not clear what type of changes have been performed and still my major concerns seem valid, the authors need to respond comment by comment:

In this rather short clinical paper, the authors present a single case of a 70 yrs old female patient with vaginal melanoma diagnosed in June 2020. The initial histopathological exam suggested undifferentiated carcinoma and the immunohistochemical tests showed that tumor cells were diffusely positive for S100, while HMB45 and p16 were positive with moderate intensity in the tumor cells, p63 and CK5/6 were negative in the tumor, and the Ki-67 labeling index was 85 approximately 60%. These are the molecular patterns nit really analyzed in the text. This is overal a potentially useful clinical application of combined immunotherapy and radiation therapy but the work needs major revisions. The abstract for example , conclusions and results are really non straight forward and highly informative. Details are missing. One reads and does not really understand which was the exact case and if the patient and other patients can really benefit from it. 

Materials and methods: very weak and without details. Radiation treatment details and the patient response needs to be better documented. 

In addition, a serious discussion must be made based on current knowledge incorporating papers documenting the role of the immune system in defining the overall outcome of radiation therapy :

1. Ventura, J.; Lobachevsky, P.N.; Palazzolo, J.S.; Forrester, H.; Haynes, N.M.; Ivashkevich, A.; Stevenson, A.W.; Hall, C.J.; Ntargaras, A.; Kotsaris, V., et al. Localized Synchrotron Irradiation of Mouse Skin Induces Persistent Systemic Genotoxic and Immune Responses. Cancer Res 2017, 77, 6389-6399, doi:10.1158/0008-5472.CAN-17-1066.

2. Rodriguez-Ruiz, M.E.; Vanpouille-Box, C.; Melero, I.; Formenti, S.C.; Demaria, S. Immunological Mechanisms Responsible for Radiation-Induced Abscopal Effect. Trends Immunol 2018, 39, 644-655, doi:10.1016/j.it.2018.06.001.

Author Response

We would like to thank the first reviewer for his valuable ideas in correcting this manuscript.

Thank you for the appreciation of our work.

According to your suggestions, we corrected the mistakes, we have completed the proposed data regarding abscopal responses based on the activation of IFN-I via cGAS and STING pathway in the irradiated cancer cells and the radiotherapy-induced bystander effect.

We hope you consider the article based on current knowledge, documenting the immune system's role in defining the overall outcome of radiation therapy combined with immunotherapy.

Best Regards,

Gabriela Stoleriu

This manuscript is a resubmission of an earlier submission. The following is a list of the peer review reports and author responses from that submission.